# Prevalence and Predictors of Depression in Women with Osteoarthritis: Cross-Sectional Analysis of Nationally Representative Survey Data

**DOI:** 10.3390/healthcare12050502

**Published:** 2024-02-20

**Authors:** Ananya Ravi, Elisabeth C. DeMarco, Sarah Gebauer, Michael P. Poirier, Leslie J. Hinyard

**Affiliations:** 1Saint Louis University School of Medicine, Saint Louis, MO 63104, USA; 2Department of Health and Clinical Outcomes Research, Saint Louis University School of Medicine, Saint Louis, MO 63104, USA; elisabeth.demarco@health.slu.edu (E.C.D.); michael.poirier@slu.edu (M.P.P.); leslie.hinyard@health.slu.edu (L.J.H.); 3Advanced Health Data Institute, Saint Louis University School of Medicine, Saint Louis, MO 63104, USA; sarah.gebauer@health.slu.edu; 4Department of Family and Community Medicine, Saint Louis University School of Medicine, Saint Louis, MO 63104, USA

**Keywords:** osteoarthritis, depression, women, aging

## Abstract

Osteoarthritis (OA) is the most common joint disease in the US and can increase the risk of depression. Both depression and OA disproportionately affect women, yet this study is one of few on depression prevalence, treatment across age groups, and predictors in women with OA. Data were extracted from the 2011–March 2020 National Health and Nutrition Examination Survey (NHANES). Women aged ≥ 45 years with self-reported osteo- or degenerative arthritis were included. Outcomes were depression (assessed with PHQ-9) and treatment (self-reported pharmacotherapy and mental health services). Logistic regression was used to examine associations between age group, covariates, and outcomes. Overall, depression prevalence was 8%, with higher proportions among those 45–64 years old. Aging was associated with reduced odds of depression (Age 65–79: OR 0.68 (95% CI: 0.52–0.89); Age 80+: OR 0.49 (95% CI: 0.33–0.74); vs. Age 45–54). Of those with a positive depression screen, 21.6% documented some form of treatment. Age group was not statistically different between those treated and those not treated. Women aged 45–64 with osteoarthritis may be at increased risk of depression, and most are not treated. As depression is related to increased pain and risk of rehospitalization, future research should prioritize interventions to increase uptake of depression treatment.

## 1. Introduction

Osteoarthritis (OA) is the most common joint disease in the United States, affecting an estimated 32.5 million US adults [1], with a predicted increase in prevalence globally with the aging of populations [2,3]. OA is the leading cause of disability in adults older than 60 and disproportionately affects women compared to men. Additionally, OA has a high economic burden [2,4,5]. In 2019, the average per-person lost direct earning was USD 189, while healthcare costs amounted to USD 1778 among working adults with OA [6]. Unfortunately, this is expected to increase with a rise in the OA prevalence and costs associated with lost wages and increased needs for informal caregiving [6,7,8].

People with OA have a higher risk of developing several comorbidities compared to the general population, including mental health disorders such as depression and anxiety [9,10]. Furthermore, aging is independently associated with both OA [11] and depression, highlighting the need to understand the burden of OA across age groups. Previous work has demonstrated that increased OA severity is associated with an increased likelihood of developing a depressed or anxious mood [12]. Additionally, a recent randomized control trial demonstrated that women with OA have a higher risk of depression compared to men with OA [9]. OA has also been associated with increased pain perception and restrictive joint movements, which may limit activities of daily living and reduce the overall mood of an individual [9]. While it is hypothesized that OA can lead to depressive symptoms, depression may also impact OA outcomes [13], including increased pain and higher risk of readmission after total joint arthroplasty [14]. 

Treatment for depression may moderate the relationship between quality of life and OA prognosis in these patients [15]. Treatment for depression can include pharmacologic management with antidepressants, non-pharmacologic methods such as establishing psychosocial therapy with a mental health professional, or a combination of these methods [16,17]. As factors related to the OA disease process including aging and chronic pain have been shown to influence the rate of utilization of depression treatments [15], there is a need to untangle this relationship, especially among women [18]. 

Prior work reported that elderly adults with OA were more likely to receive antidepressants for their depression when compared to younger adults; however, the difference in the rate of utilization of pharmacological, non-pharmacological, and combination therapies between age subgroups of women with OA has not yet been studied [18]. The physiologic aging process is influenced by sex differences, and the unique factors such as menopause and the withdrawal of estrogen in women can negatively affect the loss of bone mineral density and accelerate the development of osteoporosis compared to men [19]. These biological sex differences have a significant impact in age-related diseases, such as OA. In the era of individualized medicine, the need for sex-specific estimates remains high as many clinical studies on OA are centered on men [19]. Most affected people with OA are women [2], yet there is a paucity of studies focused on female clinical subjects.

To identify areas of intervention for women with OA, it is crucial to understand the role of aging and its effect on depression prevalence and treatment utilization among women with OA. To our knowledge, there are no studies exploring depression prevalence and treatment across age groups in a nationally representative sample of women with OA. While other studies have addressed pain severity in OA [20] and explored differences in depression manifestation between fibromyalgia and OA [21], the present article focuses solely on depression among women with OA. Identifying individuals with OA with higher prevalence of depression might illuminate vulnerable subgroups that are at risk of worsening OA outcomes. Given that depression has been previously associated with worse OA prognosis in later life, identifying subgroups with a lower rate of utilization is vital to developing tools for better treatment management [22]. 

Therefore, the aims of this study were to examine depression prevalence and depression treatment (both pharmacologic and nonpharmacologic) across age subgroups in women with OA and identify predictors of depressive symptoms and receipt of treatment. 

## 2. Methods

### 2.1. Data Source

The National Health and Nutrition Examination Survey (NHANES) is a cross-sectional survey of health behaviors, risk factors for illness, and current health status [23]. Weighted NHANES data are designed to represent the civilian, non-institutionalized United States population and are typically released in 2-year cycles. Data for the present study were drawn from both household interviews and mobile examination. This study used the past 4 cycles of available data (2011–2012, 2013–2014, 2015–2016, 2017–March 2020). This period was selected to provide an adequate sample size and avoid the confounding effects of the COVID-19 pandemic, during which the time rates of mental illness (including depression) surged nationwide [24].

### 2.2. Sample

This present analysis was limited to women who were 45 years of age or older at the time of interview and self-reported a diagnosis of “osteoarthritis or degenerative arthritis” (unweighted n = 15,399). The sample was limited to those with complete data for all measures of interest, following the convention of the svyglm package. 

### 2.3. Measures

The primary outcomes of this study were depression symptoms and receipt of depression treatment. Depression symptoms were measured using the Patient Health Questionnaire-9 (PHQ-9), where a total score ≥ 10 indicated a positive depression screen [25]. At the time of survey, participants responding affirmatively to having suicidal ideation (“Over the last 2 weeks, how often have you been bothered by the following problems: Thoughts that you would be better off dead or of hurting yourself in some way?”) or displaying signs of emotional distress upon viewing this question were assessed by the Mobile Examination Center staff physician and referred for mental health services as needed [26]. Depression treatment was examined only among those with a positive depression screen (PHQ-9 ≥ 10) and was a composite measure of both non-pharmacologic and pharmacologic treatments. Non-pharmacologic treatment was measured via a self-reported visit to a mental health professional in the last 12 months. Pharmacologic treatment was derived from review of medications, where individuals were considered treated if they reported at least one of the eligible medications (Appendix A) and the medication class recorded (SSRI, SNRI, MAO-I, TCA, Other). A composite treatment variable was created, where individuals reporting both a mental health visit and at least one medication were considered to have received both pharmacologic and non-pharmacologic treatments. Participants documenting neither pharmacologic nor non-pharmacologic treatments were considered untreated.

Age group was the primary predictor and was derived from self-reported age. Participants were grouped into age categories using 10-year increments, up to age 80. NHANES classifies any individual 80 years of age or older as “80” to protect participant anonymity, thus creating an upper limit for this analysis. 

Pain medication was measured through a self-report of eligible medications (Appendix A). Report of pain medication was categorized as non-steroidal anti-inflammatory drugs (NSAIDs) only, opioid only, both NSAID and opioid, or none. Weight status was determined through a self-report (“Has a doctor or other health professional ever told you that you were overweight?”). Race, ethnicity, health insurance, marital status, and education level were self-reported. An individual was deemed to have a gap in health insurance based on their report of lacking health insurance at any point in the last 12 months. Poverty index was calculated through the NHANES using the family income-to-federal poverty ratio and categorized as <1.85, between 1.85 and 1.3, and ≥1.3.

### 2.4. Statistical Analysis

NHANES analytic guidelines were used to account for the complex survey design and guide sample weighting in all analyses to best represent the subset of interest. Differences in age group distribution and covariates by depression status were examined with bivariate analysis using chi-square tests for categorical variables and t-test for continuous variables, all of which were approximately normally distributed. Bivariate logistic regression was used to predict the likelihood of a positive depression screen based on age category, controlling for all covariates that were significantly different (*p* < 0.05) in bivariate analysis. Results of the logistic regression are presented as odds ratio (OR) and a 95% confidence interval (95% CI). Alpha was set at 0.05.

Depression treatment was examined among all individuals with a positive depression screen. Differences in age group distribution and covariates by depression treatment status were examined with bivariate analysis using chi-square tests for categorical variables and *t*-test for continuous variables, all of which were approximately normally distributed. Regression was not conducted as there were no differences in receipt of treatment by age group, the primary exposure variable for this study.

All analyses were conducted in R [27] using RStudio (Version 4.0.0) and the tidyverse [28], RHNANES [29], tableone [30], survey [31], lmtest [32], and fastDummies [33] packages.

## 3. Results

The final analytic sample included 15,399 women with a mean age of 61 (SD: 10.4). Most participants (66.7%) self-identified as non-Hispanic white (Table 1). Overall, those with depression reported a higher prevalence of some type of pain medication use (32.6% vs. 12%) and lack of health insurance in the past year (7.2% vs. 3.6%) compared to women with OA who were not depressed (Table 1).

In an adjusted regression analysis for odds of a positive depression screen, those in the 65+ age group were less likely to screen positive for depression using PHQ-9 compared to women with OA of ages 45–54 (Figure 1, Appendix A). Those identifying as white, divorced or widowed, overweight, or using NSAIDs for pain control (as compared to no pain management) were more likely to report a positive depression screen (Figure 1, Appendix A). 

Among those with positive depression screens, there was no significant difference in receipt of treatment for older individuals with OA based on age group (Table 2). Among those who received treatment for depression, more individuals reported use of at least one pain medication (Table 2). Meanwhile, 25.4% of women with OA who received treatment for depression were taking NSAIDS, opioids, or a combination of both, compared to 11.3% of those who did not report treatment for depression (Table 2).

Among those with a positive depression screen, 61.6% documented antidepressants only, 17.4% documented only a mental health visit, and 20.9% documented both (Table 3). SSRIs were the most common antidepressant reported for those taking medication (57.4%; Table 3). 

## 4. Discussion

In this cross-sectional study of older US women with OA, we found that increased age decreased the likelihood of a positive depression screen using PHQ-9 but did not appear to be associated with receipt of treatment. Notably, seeking adequate treatment for depression can improve disease burden, including reduced pain and reduced 90-day readmission after surgery, which may be particularly pertinent for women with OA given the increased risk of comorbid depression [14]. Compared to women with OA of ages 45–54, women older than 65 were less likely to screen positive for depression (Figure 1). This finding appears to run contrary to other studies, which noted that aging has been independently associated with increased risk of both depression and OA [11,34]. 

Several possible interpretations may serve to support these findings—psychometric properties of PHQ-9, decreases in quality of life associated with activity restriction, and increased risk of depression around the time of menopause. The dimensions assessed through PHQ-9 were originally studied in an obstetrics–gynecology sample of younger patients with fewer medical comorbidities [25]. These psychometric properties have since been assessed in many populations, with controversy surrounding use of the PHQ-9 in the assessment of depression in the elderly and its applicability to those with multimorbidity, such as in a 2013 assessment of these properties in people with OA [35]. The findings of a decreased likelihood of positive depression screen with age should be interpreted in the context of the limitations of using the PHQ-9 to measure the true prevalence of depression in an older, more chronically ill population. Screening tools such as Beck Depression Inventory (BDI) and the Geriatric Depression Scale (GDS) contain items that assess for somatic symptoms of depression that may be more sensitive and specific to the sample of older women with OA in our study [35]. Moreover, PHQ-9 functions as a depression screening and may not identify people who are being treated for depression because their symptoms are adequately managed. Our study’s assessment of the prevalence of depression in older women may have been affected as a result. Although NHANES only provided PHQ-9 data, future work may elucidate the effectiveness of depression screening tools in our sample by replicating the design using BDI or GDS and observing for a variance in results. 

Additionally, the development of osteoarthritis and restriction in activity have been correlated with reduced quality of life [36,37] and depression [37,38], particularly in younger cohorts [39,40]. Many of these models, however, adjust for age and do not report differences across age groups. Reporting such age-specific results may allow for more tailored interventions. It is possible that younger women experience a greater reduction in activity with osteoarthritis compared to older women and, thus, suffer from increased depressive symptoms. A trajectory analysis demonstrated that those developing poor outcomes more often displayed greater activity limitation over time and were more often younger individuals with a greater comorbidity burden [39]. Furthermore, while those with a trajectory of an improved quality of life tended to be older and experience fewer depressive symptoms, those with a low quality of life (more often younger and with a greater depressive symptom burden) tended to maintain a low quality of life [40]. Taken together, the increased proportion of positive depression screening among women aged 45–54 in the present study may be indicative of a subgroup with a worse overall quality of life, potentially due to activity restriction or an increased burden due to experiencing osteoarthritis at a younger age.

Alternatively, the higher risk of a positive depression screen in women younger than 65 in our sample may be associated with menopause. Studies on the menopausal transition such as the Study of Women’s Health Across the Nation (SWAN) showed that women have a 2–4-fold increase in likelihood of experiencing a major depressive episode during the menopausal and immediate post-menopausal period compared to pre-menopause or late life [41]. Hormonal changes surrounding menopause may modulate the incidence and progression of osteoarthritis, with one study citing that the highest excess risk related to female sex is seen around menopause for developing hand OA [42,43]. A correlation between menopause and increased incidence [44] and severity [45] of depressive symptoms has also been observed, while the adoption of healthy lifestyle habits can mediate the risk of menopause-associated depression [46]. Although our current study was unable to measure menopause due to lack of a reliable assessment in NHANES, this transition is a vulnerable period which has strong associations with mood, showing the need for nationally representative health surveys to include this information with future data collection. 

Although the recent literature has shown that older adults are more hesitant to seek professional interventions for depression [47], we found no differences in the receipt of treatment for women with OA across age groups. Among older adults, factors such as stigma and reliance on self-management strategies can influence the perceptions of help-seeking for depression and potentially dissuade them from seeking treatment [47,48,49]. Interestingly, of the participants who received treatment for depression, a majority (61.6%) reported antidepressant use only (Table 3). A total of 17.4% documented only a mental health visit, while 20.9% documented both a mental health visit and at least one medication. SSRIs were the most common antidepressant reported for those taking medication (57.4%; Table 3). Similar to the general US population, women with OA also have a preference for pharmacologic modalities for depression treatment. Both medication and psychotherapy have important benefits for depression remission, with a combination of modalities recommended for those with moderate to severe symptoms.

Our data additionally revealed that obesity and the use of pain medication were more common amongst those with depression in our population compared to those without depression. Obesity plays a considerable role beyond its structural impact on osteoarthritis [50], including conferring increased risk of depression and changing the trajectory of depression in individuals with knee OA, such as contributing to its progression to worse depressive symptoms or the development of comorbidities [51]. The reported opioid use in women with OA was 17% in those who were depressed compared to 5.2% in those without depression (Table 1). Similarly, NSAID use was also higher in those who screened positive for depression (10.3% vs. 5.3%; Table 1). This evidence is consistent with that of Fonseca-Rodriguez et al. [52], who highlighted a significant correlation between pain, adequate pain treatment, and the severity of depression in OA patients. Understanding these factors highlights a need for the holistic management of depression and routine evaluation of obesity and adequate pain management by clinicians managing patients with OA. 

## 5. Limitations and Recommendations for Future Research Directions

This study has many strengths. This study offers a novel description of relationships between depression prevalence and age groups in women with OA. The inclusion of five cycles of national survey data from NHANES allowed for a substantially large sample size, increasing power. The weighted sample used is generalizable to the community-dwelling US population. Finally, we included a robust set of covariates for OA including self-reported obesity, a known risk factor for OA and depression, and self-reported pain medication use, of which prescription opioid use has been shown to increase the risk of MDD [12,53]. 

Importantly, our findings should be interpreted in the context of the study’s limitations. The cross-sectional nature of these data limits implications for causality and could be strengthened by following a longitudinal cohort and with discernable age groups beyond the age of 80. The age reported by NHANES classifies adults older than 80 into one group, which limits the capacity to determine the prevalence of depression in women with OA late in life. These data also represent self-report, which may be subject to recall or social desirability bias. This study defined depression based on participants screening positive for symptoms on PHQ-9. Individuals who have depression treated to remission would not be represented. Thus, the reported prevalence, risk of positive depression screen, and treatment utilization proportions should be reassessed in future work that also includes those who have been adequately treated for depression. Further data collection using tools such as GDS and BDI, which account for nuances in the somatic pain symptoms of OA, including survey questions regarding menopausal symptoms, may enrich the understanding of the implications of this study in clinical practice. 

## 6. Conclusions

In women with OA, the preliminary evidence found in this study showed that age was associated with a decreased likelihood of a positive depression screen but did not change the utilization of treatment for depression. By identifying limitations of the PHQ-9 tool in our population and the potential role of menopause in the increased prevalence of depression, the need for future work to study these associations across age groups in a longitudinal setting with more nuanced depression screening becomes evident. Understanding the complex interplay between depression, pain, and OA outcomes will further enrich clinical decision-making and offer insights to improve the care of those struggling with OA. 

## Figures and Tables

**Figure 1 healthcare-12-00502-f001:**
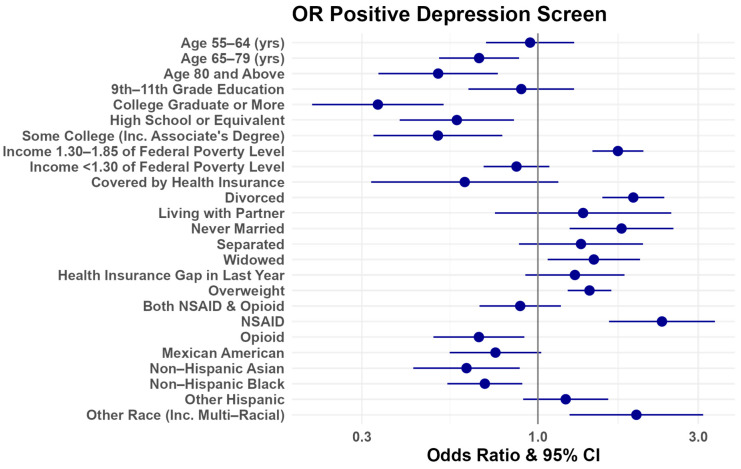
Forest plot of positive depression screen odds ratios. Data provided in Appendix A. (Created using R and R studio).

**Table 1 healthcare-12-00502-t001:** Demographics by depression screen status.

Demographics	Depressed (%)	Not Depressed (%)	*p*
**Age**			0.007
45–54	34.2	31.7	
55–64	36.6	32	
65–79	23.4	28.4	
80+	5.7	7.9	
**Race**			<0.001
Non-Hispanic White	66.7	72.8	
Non-Hispanic Black	10.7	9.9	
Non-Hispanic Asian	2.1	4.4	
Mexican American	6.8	5.3	
Other Hispanic	7.8	4.8	
Other/Multiracial	5.8	2.7	
**Citizenship**			0.543
US Citizen	94.6	95	
Not a US citizen	5.4	5	
**Marital Status**			<0.001
Married	46	65.2	
Widowed	22.2	16	
Divorced	18.6	10.7	
Separated	2.8	1.5	
Never Married	7.2	4.4	
Living with Partner	3.3	2.2	
**Lack Health Insurance Coverage**	11.8	9.1	0.006
**Gap in Insurance in Past Year**	7.2	3.6	<0.001
**Pain Medication Use**			<0.001
None	67.4	88	
NSAIDS	10.3	5.3	
Opioids	17	5.2	
Both	5.2	1.4	
**Family Income**			<0.001
>1.85 Poverty Level	45.1	70.6	
1.35 < Poverty Level ≤ 1.85	15.3	10.9	
Poverty Level ≤ 1.35	39.6	18.4	
**Education Level**			<0.001
Less than 9th grade	9.2	4.5	
9th–11th grade	15.5	8.2	
HS Graduate/GED	28	23.7	
Some college/AA	31.6	30.6	
College graduate or greater	15.6	32.9	
**Overweight**	55	41.8	<0.001
**Time Since Last Healthcare Visit**			0.956
Within 1 year	11.7	11.5	
More than 1 year	88.3	88.6	
**Misc. Analgesic Use**	0	0	-

**Table 2 healthcare-12-00502-t002:** Sample demographics by depression treatment status.

Demographics	Received Treatment for Depression (%)	No Treatment for Depression (%)	*p*
**Age**			0.966
45–54	31.8	32.3	
55–64	32.1	32	
65–79	27.6	27.4	
80+	8.5	8.3	
**Race**			<0.001
Non-Hispanic White	80	68.6	
Non-Hispanic Black	7.6	11.1	
Non-Hispanic Asian	1.4	5.8	
Mexican American	3.3	6.2	
Other Hispanic	4.3	5.5	
Other/Multiracial	3.5	2.8	
**Citizenship**			<0.001
US Citizen	97.8	93.5	
Not a US Citizen	2.2	6.5	
**Education**			0.013
Less than 9th grade	4.7	6	
9th–11th	9.5	8.9	
HS Graduate/GED	23	24.1	
Some college/AA	33.5	29.3	
College graduate+	29.4	31.7	
**Marital Status**			<0.001
Married	56.6	64.9	
Widowed	20.6	15.6	
Divorced	13.3	10.9	
Separated	1.8	1.6	
Never Married	5	4.6	
Living with Partner	2.7	2.4	
**Pain Medication Use**			<0.001
None	74.6	89.7	
NSAID	9.2	4.8	
Opioid	12.4	4.5	
Both	3.9	1	
**Overweight (Y)**	53.3	39	<0.001
**Time since Last Healthcare Visit**			0.015
Within 1 year	23.8	11	
More than 1 year	76.2	89.0	
**Health Insurance (N)**	10.5	9.5	<0.001
**Gap in Insurance in Past Year (Y)**	4.1	3.8	0.581
**Family Income**			0.001
>1.85 Poverty Level	63.8	68.7	
1.3 < poverty level < 1.85	12.4	11.1	
Poverty Level ≤ 1.35	23.8	20.1	

**Table 3 healthcare-12-00502-t003:** Use of depression treatment modalities (weighted percentage).

	(%)
**Mental Health Visit**	38.4
**Any Antidepressant Use**	82.6
**SSRI**	57.4
**SNRI**	16.4
**TCA**	7.9
**MAOI**	0.1
**Other**	12.7

## Data Availability

Data are publicly available at https://www.cdc.gov/nchs/nhanes/index.htm. (accessed on 9 July 2022) See references for further details.

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
