# Peer review of "Prevalence and Predictors of Depression in Women with Osteoarthritis: Cross-Sectional Analysis of Nationally Representative Survey Data"

_healthcare, 2024, doi:10.3390/healthcare12050502_

Round 1
Reviewer 1 Report
Comments and Suggestions for Authors
Dear, I appreciated your work and all the structural and functional elements of the manuscript are correct. On that assumption, my opinion could be positive but the ethical issue is a knot to unravel, as it is unthinkable to state that "this manuscript is not centered on human material, so we have waived the opinion of the ethics committee". In the absence of the committee's opinion, this study is not publishable, as patients are not private anyway; therefore, I urge you to obtain it and re-submit your work, as it is indeed valuable.
Additional comments:
The strengths and weaknesses: The manuscript addresses the issue of osteoarthritis (OA), which is the most common joint disease in the United States and therefore may affect depressive symptomatology in women. This study aims to identify predictors of depressive symptoms and treatment in women with OA. Data were extracted from the 2011-March 2020 National Health and Nutrition Examination Survey (NHANES), excluding patients in the Covid period as symptomatology could have generated bias. The outcomes were depression (assessed with the PHQ-9) and treatment (self-reported pharmacotherapy and mental health services). Logistic regression was used to examine associations between age groups, covariates, and outcomes. All analyses were weighted to account for the sampling design. These are all strengths, while as a weakness I confirm the ethical issue, as the sample is still human material.
Methodology: For me, the study is functional and suitable and therefore I do not see any further suggestions. By the way, the results confirm that the prevalence of depression was 8%, with a higher percentage of positive screenings for depression among subjects aged 45-64 years. Older age was associated with a reduced likelihood of depression.
The additional screening should be considered: Ethical approval I think is the real central issue.
Conclusions: The data are consistent with the discussions and conclusions, and the tables are complementary and helpful for understanding. Women aged 45-64 years with osteoarthritis may be at increased risk for depression, and although treatment of depression does not vary between age groups, most women over 45 years with symptoms of depression were not treated (thus suggesting that they should be).
References: Appropriate, nothing more needs to be added.
Author Response
Reviewer 1
Dear, I appreciated your work and all the structural and functional elements of the manuscript are correct. On that assumption, my opinion could be positive but the ethical issue is a knot to unravel, as it is unthinkable to state that "this manuscript is not centered on human material, so we have waived the opinion of the ethics committee". In the absence of the committee's opinion, this study is not publishable, as patients are not private anyway; therefore, I urge you to obtain it and re-submit your work, as it is indeed valuable.
Additional comments:
The strengths and weaknesses: The manuscript addresses the issue of osteoarthritis (OA), which is the most common joint disease in the United States and therefore may affect depressive symptomatology in women. This study aims to identify predictors of depressive symptoms and treatment in women with OA. Data were extracted from the 2011-March 2020 National Health and Nutrition Examination Survey (NHANES), excluding patients in the Covid period as symptomatology could have generated bias. The outcomes were depression (assessed with the PHQ-9) and treatment (self-reported pharmacotherapy and mental health services). Logistic regression was used to examine associations between age groups, covariates, and outcomes. All analyses were weighted to account for the sampling design. These are all strengths, while as a weakness I confirm the ethical issue, as the sample is still human material.
Methodology: For me, the study is functional and suitable and therefore I do not see any further suggestions. By the way, the results confirm that the prevalence of depression was 8%, with a higher percentage of positive screenings for depression among subjects aged 45-64 years. Older age was associated with a reduced likelihood of depression.
The additional screening should be considered: Ethical approval I think is the real central issue.
The authors appreciate Reviewer 1’s comments and would like to clarify the ethical approval statement. The manuscript lists the following:
Institutional Review Board statement: Ethical review and approval were waived for this study, as this secondary analysis of publicly available data was considered non-human subjects research.
Informed Consent Statement: Informed consent was waived, as this secondary analysis of publicly available data was considered non-human subjects research.
The determination of non-human subjects research from the Institutional Review Board is based on the use of publicly available, non-identifiable information. We did not collect any of this data nor conduct any intervention. As such, no formal IRB approval was needed. Informed consent was obtained at the time of primary data collection by staff affiliated with the National Health and Nutrition Examination Survey (NHANES), the source of the data used here. The formal letter of non-human-subjects research determination is attached.
Conclusions: The data are consistent with the discussions and conclusions, and the tables are complementary and helpful for understanding. Women aged 45-64 years with osteoarthritis may be at increased risk for depression, and although treatment of depression does not vary between age groups, most women over 45 years with symptoms of depression were not treated (thus suggesting that they should be).
References: Appropriate, nothing more needs to be added.
We appreciate Reviewer 1’s comments and thank them for their time.

Reviewer 2 Report
Comments and Suggestions for Authors
This is a study to determine the prevalence and predictors of depression in women with osteoarthritis based on a cross-sectional analysis of the 2011-March 2020 National Health and Nutrition Examination Survey. With 15,399 women included in the study, this examination is extensive and provides findings in an area of research that is understudied. Contrary to previous studies, this research found that an increase in age decreased the likelihood of a positive depression screen.
This is a well-devised and well-related study that provides a significant contribution to the literature. One problem relates to the discussion of the finding that women between 45-54 were more likely to be depressed than those older than 65. The authors included in their speculation regarding one of the two reasons for this result is that it could relate to menopause—rather than merely concluding it regards a reduction in quality of life. Their support for this assertion is from older research. Without current research to maintain this point of view, this statement that menopause may relate to depression is seemingly sexist.
There are few changes that need to be made to this otherwise well-written work before it can be accepted for publication.
Page by page suggested edits
1
Abstract: According to the Instructions for Authors of the journal Healthcare, “The abstract should be a total of about 200 words maximum.” The authors’ abstract of this submission is 256 words. When reducing the size of the abstract, please replace all use of “&” by “and”. As part of reducing the size of the Abstract, please remove the following words “Objectives”, “Methods”, “Results”, “Conclusions”. Change “data was” to “data were”.
Keywords should be found in the Abstract and follow their order of appearance. When the authors redo their Abstract, be sure to include keywords found in the Abstract and list those keywords as they appear.
Please provide a reference to “(Centers for Disease Control and Prevention [CDC], 2020)” in the reference list.
As it is 2024, statistics from 2008 are very out of date. Please find a reference to the current (within the last five years) cost of OA.
Please find a reference published from within the last five years to support or replace citation 7.
2
Please find a reference published from within the last five years to support or replace citations 13 and 14
The authors state that to their knowledge their study is the first to explore depression prevalence and treatment across age groups in a nationally representative sample of women with OA. Please indicate in the text the ways that this current study differs from https://doi.org/10.2147/JPR.S310368, and from https://doi.org/10.3390/ijerph19063413.
3
Given that citation 19 is to a 2001 reference, the authors are asked to explain why they chose this particular measure of depression symptoms and provide a current reference using this measure.
5
Table 2 Please line up the data for Education and for Martial Status with the appropriate demographic title. The data are lower than each of the related titles.
6
Please remove the italics from the names and citation numbers of the packages.
Given that RHNANES and Imtest are both supported by references published more than five years ago, the authors are asked, for each of the packages used, to explain why they chose them and to provide a reference for these two older packages to indicate that they have been used in similar studies within the last five years.
8
Change “less medical comorbidities” to “fewer medical comorbidities”.
In the discussion of the reference related to citation 29, the authors should note that these results were from 2013.
The paragraph suggesting that the reason for positive depression screen in women younger than 65 being associated with menopause is problematic. This is true both because the references supporting this suggestion are out of date and because a simpler and nonsexist explanation is that that these woman generally are more active than the older women and, as a result, their OA reduces their quality of life to a greater extent than the older women and their depression relates to this decrease in quality of life. Please refer to https://doi.org/10.1111/ggi.13879 and to https://doi.org/10.13066/kspm.2020.15.3.109 in this regard.
Comments on the Quality of English Language
There is very little to correct regarding the English. The two changes requested are made in the Comments and Suggestions for Authors.
Author Response
Reviewer 2
This is a study to determine the prevalence and predictors of depression in women with osteoarthritis based on a cross-sectional analysis of the 2011-March 2020 National Health and Nutrition Examination Survey. With 15,399 women included in the study, this examination is extensive and provides findings in an area of research that is understudied. Contrary to previous studies, this research found that an increase in age decreased the likelihood of a positive depression screen.
This is a well-devised and well-related study that provides a significant contribution to the literature. One problem relates to the discussion of the finding that women between 45-54 were more likely to be depressed than those older than 65. The authors included in their speculation regarding one of the two reasons for this result is that it could relate to menopause—rather than merely concluding it regards a reduction in quality of life. Their support for this assertion is from older research. Without current research to maintain this point of view, this statement that menopause may relate to depression is seemingly sexist.
There are few changes that need to be made to this otherwise well-written work before it can be accepted for publication.
Response: We appreciate Reviewer 2’s discussion of our paper’s findings and have addressed the suggested edits in turn below.
Page by page suggested edits
1
Abstract: According to the Instructions for Authors of the journal Healthcare, “The abstract should be a total of about 200 words maximum.” The authors’ abstract of this submission is 256 words. When reducing the size of the abstract, please replace all use of “&” by “and”. As part of reducing the size of the Abstract, please remove the following words “Objectives”, “Methods”, “Results”, “Conclusions”. Change “data was” to “data were”.
Keywords should be found in the Abstract and follow their order of appearance. When the authors redo their Abstract, be sure to include keywords found in the Abstract and list those keywords as they appear.
Response: The suggested changes were implemented and the abstract word count is now 199. Keywords appear in the abstract and are listed in order of appearance.
Please provide a reference to “(Centers for Disease Control and Prevention [CDC], 2020)” in the reference list.
Response: This reference was provided and is now reference [1].
As it is 2024, statistics from 2008 are very out of date. Please find a reference to the current (within the last five years) cost of OA.
Response: An updated reference was provided (see reference [6], published in 2019). This supports the original reference which explicitly discusses the additional cost of informal caregiving.
Please find a reference published from within the last five years to support or replace citation 7.
Response: An additional reference was provided (see reference [10], published in 2019).
2
Please find a reference published from within the last five years to support or replace citations 13 and 14
Response: Additional references were provided (see reference [17], published in 2021, and reference [15], published in 2022).
The authors state that to their knowledge their study is the first to explore depression prevalence and treatment across age groups in a nationally representative sample of women with OA. Please indicate in the text the ways that this current study differs from https://doi.org/10.2147/JPR.S310368, and from https://doi.org/10.3390/ijerph19063413.
Response: We appreciate Reviewer 2’s direction to these articles. The text has been updated (reproduced below) to specifically address these two references. Neither paper examines the same research question as the present analysis.
“While other studies have addressed pain severity in OA [1] and explored differences in depression manifestation between fibromyalgia and OA [2], the present article focuses solely on depression among women with OA. “ (note: reference numbers updated here and correspond to reference 20 and 21 respectively in the manuscript).
Given that citation 19 is to a 2001 reference, the authors are asked to explain why they chose this particular measure of depression symptoms and provide a current reference using this measure.
Response: The Patient Health Questionnaire-9 is the depression screener chosen by the Centers for Disease Control to include as part of the National Health and Nutrition Examination Survey (NHANES). As such, the original instrument is cited to refer readers to the source of this screener and information regarding the development of cutoffs to indicate depression severity. No changes were made in the text. This instrument remains widely used today – see the following:
Hudgens S, Floden L, Blackowicz M, Jamieson C, Popova V, Fedgchin M, Drevets WC, Cooper K, Lane R, Singh J. Meaningful Change in Depression Symptoms Assessed with the Patient Health Questionnaire (PHQ-9) and Montgomery-Åsberg Depression Rating Scale (MADRS) Among Patients with Treatment Resistant Depression in Two, Randomized, Double-blind, Active-controlled Trials of Esketamine Nasal Spray Combined With a New Oral Antidepressant. J Affect Disord. 2021 Feb 15;281:767-775. doi: 10.1016/j.jad.2020.11.066. Epub 2020 Nov 14. PMID: 33261932.
Costantini L, Pasquarella C, Odone A, Colucci ME, Costanza A, Serafini G, Aguglia A, Belvederi Murri M, Brakoulias V, Amore M, Ghaemi SN, Amerio A. Screening for depression in primary care with Patient Health Questionnaire-9 (PHQ-9): A systematic review. J Affect Disord. 2021 Jan 15;279:473-483. doi: 10.1016/j.jad.2020.09.131. Epub 2020 Oct 6. PMID: 33126078.
5
Table 2 Please line up the data for Education and for Martial Status with the appropriate demographic title. The data are lower than each of the related titles.
Response: The table was revised to correct the misalignment.
6
Please remove the italics from the names and citation numbers of the packages.
Response: While the italics were removed, the citation numbers were retained. These packages are software and resources which the authors did not create. As such, the citations are critical to give appropriate credit to the developers and direct readers to additional information regarding these tools.
Given that RHNANES and Imtest are both supported by references published more than five years ago, the authors are asked, for each of the packages used, to explain why they chose them and to provide a reference for these two older packages to indicate that they have been used in similar studies within the last five years.
Response: The references indicated are more than 5 years old because these packages were created more than 5 years ago. The papers cited are the original documentation for these packages – updated documentation is available in CRAN and github, which can be found by reviewing the original documentation cited. RNHANES contains specific functions that facilitate importing data from the NHANES website and ease data management. lmtest was used to conduct in-depth analysis of the final regression models and to ensure modeling assumptions were met. These packages continue to be used in contemporary studies, such as the following:
RNHANES
- Dodson, Robin E., et al. "Consumer behavior and exposure to parabens, bisphenols, triclosan, dichlorophenols, and benzophenone-3: Results from a crowdsourced biomonitoring study." International journal of hygiene and environmental health 230 (2020): 113624.
IMtest
- Amengual D, Fiorentini G, Sentana E. Multivariate Hermite polynomials and information matrix tests. Econometrics and Statistics. January 2024. doi:10.1016/j.ecosta.2024.01.005
- Prokhorov A, Schepsmeier U, Zhu Y. Generalized information matrix tests for copulas. Econometric Reviews. 2019;38(9):1024-1054. doi:10.1080/07474938.2018.1514023
- Politis, Ioannis, et al. "COVID-19 lockdown measures and travel behavior: The case of Thessaloniki, Greece." Transportation Research Interdisciplinary Perspectives 10 (2021): 100345.
- Stevens, Matthew Leigh, et al. "Cardiorespiratory fitness, occupational aerobic workload and age: workplace measurements among blue-collar workers." International Archives of Occupational and Environmental Health 94 (2021): 503-513.
8
Change “less medical comorbidities” to “fewer medical comorbidities”.
Response: This change was implemented.
In the discussion of the reference related to citation 29, the authors should note that these results were from 2013.
Response: This has been noted in the text.
The paragraph suggesting that the reason for positive depression screen in women younger than 65 being associated with menopause is problematic. This is true both because the references supporting this suggestion are out of date and because a simpler and nonsexist explanation is that that these woman generally are more active than the older women and, as a result, their OA reduces their quality of life to a greater extent than the older women and their depression relates to this decrease in quality of life. Please refer to https://doi.org/10.1111/ggi.13879 and to https://doi.org/10.13066/kspm.2020.15.3.109 in this regard.
Response: We appreciate Reviewer 2’s comment drawing the impact of activity restriction and quality of life on depressive symptoms among younger women with osteoarthritis. We have incorporated a paragraph to this effect in the manuscript to enhance the discussion (reproduced below). Additionally, more recent references were added to support our discussion of menopause-associated depression[3,4]. Depression risk does appear increased during the menopause transition and as such we feel it is relevant to the discussion of the present results.
Additional text:
Additionally, development of osteoarthritis and restriction in activity have been correlated with reduced quality of life[36,37] and depression[37,38], particularly in younger cohorts [39,40]. Many of these models, however, adjust for age and do not report differences across age groups. Reporting such age-specific results may allow for more tailored interventions. It is possible that younger women experience a greater reduction in activity with osteoarthritis compared to older women and thus suffer from increased depressive symptoms. Trajectory analysis demonstrated that those developing poor outcomes more often displayed greater activity limitation over time and were more often younger individuals with greater comorbidity burden [39]. Furthermore, while those with a trajectory of improved quality of life tended to be older and experience fewer depressive symptoms, those with low quality of life (more often younger and with greater depressive symptom burden) tended to maintain a low quality of life[40]. Taken together, the increased proportion of positive depression screening among the women ages 45-54 in the present study may be indicative of a subgroup with worse overall quality of life, potentially due to activity restriction or increased burden due to experiencing osteoarthritis at a younger age.
Round 2
Reviewer 1 Report
Comments and Suggestions for Authors
The revisions made are functional for publication and therefore I give my positive opinion.